# Preclinical Study of *Plasmodium* Immunotherapy Combined with Radiotherapy for Solid Tumors

**DOI:** 10.3390/cells11223600

**Published:** 2022-11-14

**Authors:** Zhu Tao, Wenting Ding, Zhipeng Cheng, Yinfang Feng, Zhongkui Kang, Runmin Qiu, Siting Zhao, Wen Hu, Fang Zhou, Donghai Wu, Ziyuan Duan, Li Qin, Xiaoping Chen

**Affiliations:** 1School of Life Sciences, Division of Life Sciences and Medicine, University of Science and Technology of China, Hefei 230027, China; 2State Key Laboratory of Respiratory Disease, Center for Infection and Immunity, Guangzhou Institutes of Biomedicine and Health, Chinese Academy of Sciences, Guangzhou 510530, China; 3CAS Lamvac (Guangzhou) Biomedical Technology Co., Ltd., Guangzhou 510663, China

**Keywords:** *Plasmodium* immunotherapy, radiotherapy, glioma, lung cancer, antitumor immune response

## Abstract

Immune checkpoint blockade therapy (ICB) is ineffective against cold tumors and, although it is effective against some hot tumors, drug resistance can occur. We have developed a *Plasmodium* immunotherapy (PI) that can overcome these shortcomings. However, the specific killing effect of PI on tumor cells is relatively weak. Radiotherapy (RT) is known to have strong specific lethality to tumor cells. Therefore, we hypothesized that PI combined with RT could produce synergistic antitumor effects. We tested our hypothesis using orthotopic and subcutaneous models of mouse glioma (GL261, a cold tumor) and a subcutaneous model of mouse non-small cell lung cancer (NSCLC, LLC, a hot tumor). Our results showed that, compared with each monotherapy, the combination therapy more significantly inhibited tumor growth and extended the life span of tumor-bearing mice. More importantly, the combination therapy could cure approximately 70 percent of glioma. By analyzing the immune profile of the tumor tissues, we found that the combination therapy was more effective in upregulating the perforin-expressing effector CD8^+^ T cells and downregulating the myeloid-derived suppressor cells (MDSCs), and was thus more effective in the treatment of cancer. The clinical transformation of PI combined with RT in the treatment of solid tumors, especially glioma, is worthy of expectation.

## 1. Introduction

Cancer is one of the leading causes of death worldwide, with almost 10 million people dying of cancer in 2020 [1,2]. Solid tumors account for the majority of all tumors, while hematological tumors account for only a small portion [1,2]. Cancer immunotherapy is the fastest growing new approach for cancer treatment and has made great advances in the past two decades; in particular, with the discoveries of immune checkpoint blockade therapy (ICB) for solid tumors and chimeric antigen receptor (CAR) T cell therapy for hematologic tumors [3,4]. ICB, such as PD-1 antibody, removes the inhibitory signal of effector T cells by blocking the binding of inhibitory receptor (such as PD-1) on the surface of T cells and inhibitory ligand (such as PD-L1) on the surface of tumor cells, enabling T cells to kill tumor cells [5]. A large number of clinical trials and clinical practices have proven that ICB is effective against many types of solid tumors [6], but elicits relatively low response rate in most tumors [7], possibly due to their heterogeneity. Although solid tumors are heterogeneous and complex in nature [8], they can be roughly classified into immune hot (T-cell inflamed) and cold (non-T-cell inflamed) tumors [9,10]. ICB is usually only effective in hot tumors and is proven non-efficacious for cold tumors as ICB acts by removing the inhibitory signals from tumor infiltrating and residential effector T cells that have been activated and then suppressed by tumor cells, but it would not work if there are no or few T cells inside the tumor [7,11]. This is a major drawback for greater applications of ICB in the treatment of cold tumors. The current strategy to overcome this problem is to combine ICB with other anticancer therapies in order to turn cold tumors into hot ones [12].

Through over a decade of research in mouse tumor models, we have developed a new type of immunotherapy that works in a different way to ICB; namely, *Plasmodium* immunotherapy (PI), which uses the infection of a benign form of *Plasmodium* parasite to activate the immune system for the treatment of malignant solid tumors [13]. PI first activates the innate immunity, including the activation of dendritic cells (DCs), macrophages, and natural killer (NK) cells, which release large amounts of proinflammatory/Th1-type cytokines (including IL-12, TNF-α, IFN-γ) [14,15] that promote the infiltration of immune cells (CD45^+^ leukocytes, including innate immune cells and effector T cells) into the tumor tissues [15]. In this way, the activated innate immune cells kill some of the tumor cells, and then the dying tumor cells release tumor antigens or tumor-associated antigens that activate the effector T cells [15]. Through some undefined mechanisms, PI down-regulates the expression level of PD-1 on CD8^+^ T cells within tumor tissues [16] so that these effector T cells can effectively kill tumor cells, whilst the expression level of the PD-1 on CD8^+^ T cells in peripheral blood is up-regulated [17]. At the same time, PI can significantly down-regulate the number and function of immune suppressor cells, such as myeloid-derived suppressor cells (MDSCs), regulatory T cells (Tregs), tumor-associated macrophages (TAMs) and cancer-associated fibroblasts (CAFs) in tumor tissues, and therefore systematically remove the immunosuppressive tumor microenvironment [13,16,18]. PI also inhibits tumor angiogenesis through multiple pathways and targets [18,19,20], as well as inhibiting the epithelial-mesenchymal transition (EMT) of tumor cells by suppressing the CCR10-mediated PI3K/Akt/GSK-3β/Snail signaling pathway. PI therefore inhibits the growth and metastasis of solid tumors in mice, prevents tumor recurrence after surgical resection [21], and significantly prolongs the survival time of tumor-bearing mice [13]. Based on these studies, clinical trials of PI for advanced solid tumors have been approved and are ongoing in China (NCT02786589, NCT03474822 and NCT03375983) [13]. However, PI also has an obvious insufficiency; that is, the specific killing effect of tumor cells is relatively weak and needs to be combined with existing therapies to further enhance its efficacy.

Radiotherapy (RT) is a first line treatment for many primary and metastatic tumors, with 40% to 60% of cancer patients receiving RT for curative and palliative purposes [22,23]. RT directly kills cancer cells by damaging cellular DNA, thereby releasing tumor-associated antigens and inducing immunogenic cancer cell death, which in turn triggers a powerful antitumor innate and adaptive immune response [24]. RT also potentially induces the modulation of signal transduction and alteration in the inflammatory tumor microenvironment, such as the induction of chemokines and vascular adhesion molecules that promote the tumor infiltration of effector T cells [24]. However, RT alone is inadequate to completely kill all tumor cells in some cancers and has a high risk of cancer metastasis and recurrence [25]. Abscopal effects of RT are rare and dependent on the presence of T cells that can be increased by combination of radiotherapy and immunotherapy [26]. Combination therapy is one of the main strategies of antitumor therapy at present [26,27,28]. Previous studies have demonstrated that the combination of RT and ICB prolongs the survival of mice with an orthotopic brain tumor [29], lung cancer [30,31], breast cancer [28] and colon carcinoma [28,30]. Combination therapy could also convert an immune cold tumor into a hot tumor [10]. More than half of patients have indications for radiotherapy [32]. Therefore, we hypothesized that PI combined with RT could generate a robust antitumor immune response against various solid tumors.

According to the Global Cancer Statistic 2020, cancers of the brain and central nervous system account for 1.6% of newly diagnosed cancer cases and 2.5% of deaths worldwide, and glioma is the most common malignancy among primary brain tumors in adults [1]. Glioma is known as cold tumor [9,10] and has a low mutation load and rare infiltration of immune effector cells for the immunological interventions [33]. Lung cancer remains one of the most commonly diagnosed cancers (11.7% of the total cases) and was a leading cause of cancer death (18% of the total cases) in 2020 [1]. Non-small cell lung cancer (NSCLC) accounts for approximately 85% of all lung cancers [34], and is considered a hot tumor [35]. Therefore, we selected the cold glioma and the hot NSCLC models to test our hypothesis. Our results indicate that the combination of PI and RT has synergistic and complementary antitumor effects in these tumor models.

## 2. Materials and Methods

### 2.1. Mice

Six to eight weeks old C57BL/6 wild type female mice were purchased from the Vital River Experiment Animal Limited Company (Beijing, China). All mice were maintained in a specific-pathogen-free barrier facility of the Animal Center, according to the Guide for the Care and Use of Laboratory Animals, established by the Guangzhou Institutes of Biomedicine and Health, Chinese Academy of Sciences (IACUC Number: 2014010 and Date: 12 September 2014).

### 2.2. Cell Culture and Parasites

The murine glioblastoma cell line GL261 (CRL-1887) and murine Lewis lung cancer cell line (LLC, CRL-1642) were purchased from the American Type Culture Collection (ATCC). GL261/mCherry-Luciferase (GL261-Luc) cells were constructed by using the 3rd generation lentiviral system [36] to stably express the report genes of mCherry and firefly luciferase. GL261 and GL261-Luc cells were maintained in a Dulbecco’s modified Eagle’s medium (DMEM, Cat# 11995065, Invitrogen Gibco^TM^, Waltham, MA, USA), supplemented with 10% fetal bovine serum (FBS, Cat# 10270, Invitrogen Gibco^TM^) and 1% penicillin-streptomycin (Cat# 15140122, ThermoFisher, Waltham, MA, USA). LLC cells were maintained in Roswell Park Memorial Institute media 1640 (RPMI-1640, Cat# SH30809.01B, Hyclone), supplemented with 10% FBS and 1% penicillin-streptomycin. All cells were grown in a humidified atmosphere of 5% CO_2_ at 37 °C.

The murine nonlethal *Plasmodium yoelii* 17XNL (Py) strain was obtained from the Malaria Research and Reference Reagent Resource Center (MR4). Py was intraperitoneally injected into and propagated in C57BL/6J mice. Parasitemia was monitored with Giemsa staining (Cat# 48900, Sigma-Aldrich, Burlington, MA, USA) of the thin blood film and examined at a magnification of 100× oil immersion microscope. The parasitemia (%) was calculated by the number of *Plasmodium*-infected erythrocytes, counted among at least 1000 erythrocytes.

### 2.3. Tumor Models and Animal Grouping

For the subcutaneous tumor model, 5 × 10^5^ LLC or 2 × 10^6^ GL261 cells were subcutaneously (s.c.) inoculated into the right flank of C57BL/6J mice. The inoculation volume was 100 μL for each mouse.

The intracerebral inoculation method was improved [29,37]. After an intraperitoneal injection of 240 mg/kg avertin for anesthesia, the mouse head was fixed on a stereotactic apparatus (Cat# 68046, RWD, Shenzhen, China); 2 × 10^5^ GL261-Luc cells in a volume of 1 μL using a 5 μL Hamilton micro-syringe (26-gauge needle) were stereotactically injected into a 1 mm diameter drill hole in the left striatum that was defined by the following coordinates: 1.8 mm lateral to bregma, 1 mm posterior to the coronal suture, 3.4 mm deep to the cortical surface. The tumor burden was monitored by bioluminescent imaging after tumor cell inoculation.

In any murine tumor model study, such as the orthotopic GL261 model, subcutaneous GL261 model or subcutaneous LLC model, mice are divided into 4 treatment groups using stratified random sampling to prevent cage effects: (1) the Con (treated with uninfected erythrocytes as control) group; (2) the PI (treated with *Plasmodium* immunotherapy) group; (3) the RT (treated with radiotherapy) group; and (4) the PI+RT (treated with *Plasmodium* immunotherapy plus radiotherapy) group. The mice were randomly allocated based on the bioluminescent signal or tumor volume, on day 7 or 10 after tumor inoculation, so that the average bioluminescent signal or tumor volume in each group was roughly equivalent. The long and short diameter of the tumors were measured using a digital caliper. The tumor volume was calculated using the following formula: V = (ab^2^)/2, where V was the tumor volume, a was the long diameter, and b was the short diameter.

The animals were euthanized according to humane endpoints, including neurologic disturbances, hunched posture, lethargy, weight loss, inability to ambulate, large tumor volume (exceeding 1500 mm^3^), and so on.

### 2.4. Treatment Regimens

Seven days after tumor inoculation, each mouse was injected with 5 × 10^5^
*Plasmodium*-infected erythrocytes or the same number of uninfected erythrocytes. The mice that were inoculated with GL261 (or GL261-Luc) cells or LLC cells and injected with non-infected erythrocytes were used as the control group.

The mice were irradiated with an X-ray beam using the RS2000 X-ray irradiator (Rad Source Technologies, Boca Raton, FL, USA), according to the manufacturer’s instructions. Each mouse in the RT group or the PI+RT group received a single dose of 12 Gy radiation, with 0.3 mm copper sheets filtration at a rate of approximately 1.47 Gy/minute, under operating conditions of 160 kV and 25 mA. Each mouse in the intracerebral inoculation model received local radiation, which was placed inside a custom-built lead box, 2 cm thick, that exposes the head to radiation whilst shielding the body from the ears down. Each mouse in the s.c. inoculation model received local radiation and was put into the lead box mentioned above, which covered the body except the inoculation position.

### 2.5. Luciferase In Vivo Imaging

The PerkinElmer IVIS Spectrum in vivo imaging system (PerkinElmer, Waltham, MA, USA) was used to obtain the bioluminescent signal and imaging for evaluating the tumor burden in the intracerebral inoculation model. Each mouse was injected intraperitoneally with 150 mg/kg body weight D-Luciferin, and potassium salt (Cat# 40902ES03, Yeasen, Shanghai, China) as firefly luciferase’s substrate. Quantifications of the photon fluxes and images were performed using the Living Image software of the IVIS imaging Spectrum system (Version 4.5, PerkinElmer).

### 2.6. Flank Tumor Rechallenge Experiments

All of the cured mice with long-term survival in the GL261 orthotopic model or subcutaneous model were rechallenged with the subcutaneous inoculation of 2 × 10^6^ syngeneic GL261 cells in the right flank and the inoculation of 5 × 10^5^ non-syngeneic LLC cells in the left flank. Both cells were suspended in a sterile 0.9% sodium chloride solution. The inoculation volume was 100 μL for each mouse. Naïve mice were also injected with the same number of GL261 cells and LLC cells as the controls. Tumor volume was measured every 3 days.

### 2.7. Western Blotting

The tumor tissues were homogenized and lysed in RIPA lysis buffer (Cat# KGP702-100, KeyGene, Nanjing, China), containing a protease inhibitor cocktail (Cat# P1010, Beyotime, Nantong, China) and phosphatase inhibitor cocktail D (Cat# P1096, Beyotime), on ice for 30 min, and then centrifuged to collect the supernatant. Protein samples with equal quantity were separated by PAGE electrophoresis (Cat# M00665, GenScript SurePAGE^TM^, Nanjing, China), and then transferred to PVDF membranes (Cat# ISEQ00010, Millipore, Burlington, MA, USA). Following incubation with an appropriate secondary antibody, bands were detected with ECL reagents (Cat# WBULS0500, Millipore) and visualized using the Chemiluminescent Image Analyzer (Tanon 5200, Tanon Science and Technology, Shanghai, China). The densitometry and semi-quantitative of the bands’ signals were quantified using ImageJ 1.38 (NIH, https://imagej.nih.gov/ij/, accessed on 26 October 2022). Results were presented as the ratio of two target proteins’ densitometry. The following antibodies were used in this section: anti-caspase 3 (Cat# ab184787, Abcam, Cambridge, UK); anti-GAPDH antibody (Cat# ab8245, Abcam); HRP-linked goat anti-rabbit IgG H&L antibody (cat# ab97051, Abcam); HRP-linked anti-mouse IgG H&L antibody (Cat# 7076, CST, Louisville, KY, USA).

### 2.8. Single Cell Sample Preparation and Isolation

The mice were sacrificed on day 19 after tumor inoculation. For the blood samples, blood was drawn and anticoagulated with EDTA; 200 μL blood was placed into a Trucount^TM^ tube (Cat# 340334, BD Biosciences, Franklin Lakes, NJ, USA) for measuring the absolute number of leukocytes, according to the manufacturer’s instructions. Ammonium chloride potassium (ACK) lysis buffer was used to lyse the erythrocytes. The cells were washed and resuspended with phosphate buffered saline (PBS), following flow cytometry staining.

For the spleen samples, splenocytes were harvested using frosted lass slides through gently grinding the whole spleens in RPMI-1640, containing 2% FBS [38]. The splenocytes were filtered through a 70-µm nylon cell strainer (Cat# 352350, Corning Falcon^TM^, Glendale, AZ, USA), into 50 mL conical tubes (Jet Biofil, Guangzhou, China). ACK lysis buffer was used to lyse erythrocytes. Trypan blue exclusion was then used to count the cells, using a Countstar FL automatic cell counter (Ruiyu, Shanghai, China), according to the manufacturer’s instructions. The cells were washed and resuspended with PBS following flow cytometry staining.

For the tumor samples, tumors were harvested and cut into small fragments, and were then digested with Hank’s Balanced Salt Solution (HBSS), containing 1 mg/mL collagenase D (Cat# 11088866001, Roche, Basel, Switzerland), 20 U/mL deoxyribonuclease I (DNase I, Cat# D5025, Sigma-Aldrich) and 3 mM CaCl_2_, for 30~60 min, at 37 °C. ACK lysis buffer was used to lyse erythrocytes. Subsequently, all of the cells were filtered through a 70-µm nylon cell strainer (Cat# 352350, Corning Falcon^TM^), washed with 10 mL RPMI 1640 containing 2% FBS, and collected by centrifugation (300× *g*, 5 min). Pelleted cells were added with Precision Count Beads™ (Cat# 424902, Biolegend, San Diego, CA, USA) to obtain the absolute number of cells, and resuspended in PBS following flow cytometry staining.

### 2.9. Flow Cytometry Analysis

The cells were incubated in 100 μL PBS, using a Zombie yellow fixable viability kit (Cat# 423104, Biolegend) with 1:500 dilution for assessing live or dead status of cells at room temperature, in the dark for, 10 min, and were washed with 2 mL cell staining buffer (Cat# 420201, Biolegend). The Fc receptors were blocked by pre-incubating the cells with 0.25 µg of TruStain FcX™ (anti-mouse CD16/32 antibody, Cat# 101320, Biolegend) per 10^6^ cells, in 100 µL cell staining buffer, for 10 min in the dark, on ice. The cells were washed with 2 mL cell staining buffer. For surface markers staining, the cells were incubated with the appropriate dilution of fluorescence-conjugated surface antibodies (anti-CD45, Cat# 103147, Biolegend; anti-CD3, Cat# 100204, Biolegend; anti-CD4, Cat# 100538, Biolegend; anti-CD8a, Cat# 100752, Biolegend; anti-CD11b, Cat# 101261, Biolegend; anti-CD25, Cat# 102051, Biolegend; anti-Ly6C, Cat# 128032, Biolegend, and anti-Ly6G, Cat# 127641, Biolegend), for 30 min in the dark, on ice. For intracellular and nuclear staining, a FoxP3/Transcription factor staining buffer set (Cat# 00-5523, Invitrogen eBioscience^TM^, Waltham, MA, USA) was used for fixation and permeabilization procedures, according to the manufacturer’s instructions. Following the permeabilization procedures, the cells were stained with an intracellular antibody (anti-perforin, Cat# 154303, Biolegend) and nuclear antibody (anti-FoxP3, Cat# 126404, Biolegend), in 1× permeabilization buffer, for 30 min in the dark, on ice. The cells were washed, twice, using 1× permeabilization buffer. The cells were resuspended in staining buffer. The cells were filtered using a 200 mesh-filter before analysis by Cytek Aurora spectrum flow cytometry (Cytek Bioscience, Fremont, CA, USA). The data were analyzed using FlowJo software V10.6 (Tree Star Inc., https://www.flowjo.com/, accessed on 26 October 2022).

### 2.10. Immunohistochemical Analysis

At the time of sacrifice, the mouse tumor tissues were removed and fixed in 10% paraformaldehyde (Cat# ZLI-9381, ZSGB-Bio, Beijing, China), in PBS for 24 h. Antigen retrieval was performed under 0.01 mol/L sodium citrate buffer (pH 6.0), before incubation, with primary antibody to Ki67 (cat# ab16667, Abcam, Cambridge, UK) or CD3 (cat# ab16669, Abcam) and secondary antibody (HRP-linked goat anti-rabbit IgG H&L, Cat# ab97051, Abcam). Brightfield images were then captured and analyzed.

### 2.11. Statistical Analysis

The survival of the tumor-bearing mice was analyzed using the Kaplan-Meier method and compared using the Log-rank test. A two-tailed, unpaired t test was used to compare the significant differences between two independent groups. All of the data were analyzed using GraphPad Prism 9 (GraphPad Software, https://www.graphpad.com/scientific-software/prism/, accessed on 26 October 2022). Values of *p* ≤ 0.05 were considered statistically significant (*, *p* ≤ 0.05; **, *p* ≤ 0.01; and ***, *p* ≤ 0.001).

## 3. Results

### 3.1. Antitumor Effect of the Combination of Plasmodium Immunotherapy (PI) and Radiotherapy (RT) in Orthotopic Glioma Model

To observe the antitumor effect of PI on the growth of “cold” glioma GL261 in mice, we designed the mouse orthotopic intracerebral (Appendix A) and subcutaneous (Appendix A) tumor models. The increase in the bioluminescent signal of luciferase represented the tumor growth. The results showed that the increase in the bioluminescent signal of GL261-Luc (*p* = 0.03, Appendix A), in the intracerebral model, or the increase in tumor volume of GL261 in the subcutaneous model (*p* = 0.03, Appendix A) in the PI group was slower than that in the corresponding control group. The median survival time of tumor-bearing mice in the PI group, in the intracerebral tumor model, was 55.5 days, which was significantly longer than that of 40.5 days in the control group (*p* = 0.005, Appendix A); the median survival time of the mice in PI group in the subcutaneous tumor model was 41.5 days, which was significantly longer than that of 35.5 days in the control group (*p* = 0.04, Appendix A).

Next, we investigated the antitumor effect of the combination of PI and RT in the GL261 glioma model. The experimentation of the orthotopic intracerebral inoculation model was designed as described in Figure 1A. The parasitemia dynamics of Py were similar between the PI group and PI+RT group (Appendix A). The living images were shown to illustrate the growth trend of GL261-Luc cells in the brain (Figure 1B). The results showed that the tumor in the control group grew fastest, followed by the PI group, then by the RT group, and the tumor in the PI+RT group grew slowest of all, which perhaps could be more appropriately described as barely growing at all (Figure 1B,C). The median survival time in the control group, PI group, and RT group was 40 days, 48 days, and 88 days, respectively, but that in the PI+RT group was more than 210 days (Figure 1D). Unexpectedly, five of the seven mice in the PI+RT group showed long-term survival (tumor-free survival for at least 210 days, which could be termed as “cured”). These mice were further tested if they had already produced durable antitumor immune memory through a tumor rechallenge experiment. Naïve or “cured” mice were inoculated in a flank with syngeneic GL261 and in the other flank with non-syngeneic LLC cells. The result showed that all “cured” mice had a very small tumor volume of GL261 at the end of experiment; that is to say, their tumors barely grew at all. In contrast, the tumors in the naïve mice grew very fast (*p* < 0.001 between the two groups, Figure 1E). No significant difference was observed in the tumors of LLC between the naïve and “cured” mice (*p* > 0.05, Figure 1F). These data suggest that the “cured” mice obtained the tumor-specific immune memory.

### 3.2. Antitumor Effect of the Combination of PI and RT in Subcutaneous Glioma Model

We repeated the experiment in the subcutaneous GL261 glioma model (Figure 2A). The parasitemia dynamics of Py were similar between the PI group and PI+RT group (Appendix A). The results showed that the tumor in the control group grew fastest, followed by the PI group, then by the RT group, and the tumors in the PI+RT group had almost disappeared by the experiment endpoint (Figure 2B). The tumor weight on day 29 after tumor inoculation differed significantly among these groups (Figure 2C,D). The median survival rates were 41.5 days, 60 days, 92 days and >120 days in the control group, PI group, RT group and PI+RT group, respectively. Importantly, seven out of the ten mice in the PI+RT group experienced complete tumor regression at the experiment endpoint, which could be described as “cured” (Figure 2E). We further tested to determine whether the “cured” mice developed long-term immune memory; the naïve or “cured” mice were rechallenged in a flank with syngeneic GL261 cells (Figure 2F) and in the other flank with non-syngeneic LLC cells (Figure 2G). The tumor volume of non-syngeneic LLC was similar between the two groups (Figure 2G), however, the syngeneic GL261 in the “cured” mice barely grew, while the tumor in the naïve mice grew quickly (Figure 2F, *p* = 0.02). These results suggest that the “cured” mice obtained long-term tumor-specific immune memory.

### 3.3. Antitumor Effect of the Combination of PI and RT in Lung Cancer Model

We further investigated the antitumor effect of the combination of PI and RT in a “hot” lung cancer model that was subcutaneously inoculated with LLC cells. We first tested the effect of different doses of X-ray on LLC growth for selecting optimized RT dose (Appendix A). Then, we selected 12 Gy as the treatment dose. The design of the experiment is described in Figure 3A. The parasitemia dynamics of Py were similar between the PI group and PI+RT group (Appendix A). The results showed that the tumor in the control group grew fastest, which was followed by the RT group, then by the PI group, and the tumor in the PI+RT group grew slowest (Figure 3B). The tumor weight on day 19 after tumor inoculation was significantly different (Figure 3C,D). The median survival time was 28.5 days, 35 days, 34.5 days and 49 days in the control group, PI group, RT group and PI+RT group, respectively (Figure 3E).

### 3.4. The Effect of PI and RT Combination on Lung Cancer Cell Proliferation and Apoptosis

As shown in Figure 4A,B, the lung tumor tissues were stained with the antibody against the proliferation marker Ki67. The Ki67 expression level in the control group was the highest, which was followed by the RT group, then by the PI group, and the expression level in the PI+RT group was the lowest (Figure 4A,B). These results suggested that PI was more effective than RT in the inhibition of tumor cell proliferation, and the combination of both was better than each of them in isolation. Radiation-induced apoptosis was a caspases-dependent process [39,40], therefore we used caspase 3 to detect the level of cell apoptosis. The ratio of cleaved caspase 3 to pro-caspase 3 represented the apoptotic levels. The results showed that the apoptotic level in the PI+RT group was the highest, which was followed by the PI group, then by the RT group, and the apoptotic level in the control group was the lowest (Figure 4C,D). These results suggested that the combination of both treatments was better than each of them in isolation for promoting tumor cell apoptosis.

### 3.5. The Effect of PI and RT Combination on the Immune Profiles in Tumor Microenvironment

Glioma is a typical cold tumor [9,10]. The CD3^+^ T cells infiltrating into the tumor in orthotopic glioma model was observed by immunohistochemical analysis, using an anti-CD3 antibody. The PI group, RT group and PI+RT group all had higher CD3^+^ T cells infiltrating into the brain tumors compared with the control group (Figure 5A,B). The PI+RT group had the highest CD3^+^ T cells infiltration (Figure 5A,B). These results suggested that both PI and RT could transform a cold tumor into a hot tumor, and PI combined with RT had a more significant effect on such a transformation.

As the (either orthotopic or subcutaneous) GL261 glioma tumor volume in the PI+RT combination group was too small (or nonexistent) (Figure 1B,C and Figure 2B,D) to obtain enough immune cells for a series of analysis, with the exception of the CD3^+^ T cells calculation, as mentioned above, the LLC model was selected to further investigate the immune profiles in tumor microenvironment. The tumor tissues were harvested on day 19 after tumor inoculation and the single cells were separated in order to analyze the immunophenotypes. The gating strategy of flow cytometry is presented in Appendix A.

T cells, especially CD8^+^ T cells, play a key role in anticancer immunity [41]. As shown in Appendix A, a higher level (absolute number) of CD45^+^ leukocytes in the PI group (*p* = 0.04), and a lower level of these cells in the RT group (*p* = 0.04), were observed in comparison to the control group; there was no difference in the level of leukocytes between the PI+RT group and control group. The PI group had a higher CD3^+^ T/CD45^+^ cell proportion (*p* = 0.0009) and a higher CD3^+^ T cell absolute number (*p* = 0.0009); similarly, the RT group also had a higher CD3^+^ T/CD45^+^ cell proportion (*p* = 0.0012) and a higher CD3^+^ T cell absolute number (*p* = 0.04) compared to the control group (Appendix A). The PI+RT group had a higher CD3^+^ T/CD45^+^ cell proportion (Appendix A) and a higher CD3^+^ T cell absolute number (Appendix A) than the PI group and the RT group (all *p* < 0.05).

The CD8^+^ T/CD45^+^ cell proportion (Figure 6A), CD8^+^ T cell absolute number (Figure 6B), CD4^+^ T/CD45^+^ cell proportion (Appendix A), and CD4^+^ T cell absolute number (Appendix A) in the PI+RT group was the highest among all of the groups (all *p* < 0.05). The data concerning the cell number, shown in Figure 6B, Appendix A, suggests that PI was better than RT for promoting the T cells (both CD8^+^ T and CD4^+^ T) infiltration into tumors, and the combination of both was better than each single therapy. The PI group (*p* = 0.04) had a lower CD4^+^ T/CD8^+^ T cell ratio than the control group, and the RT group (*p* = 0.02) had a higher CD4^+^ T/CD8^+^ T cell ratio than the control group; there were no significant differences in this ratio between the PI+RT group and control group (Appendix A).

Perforin is one of the critical immune effector molecules secreted by activated CD8^+^ T cells that can induce tumor cells apoptosis or directly kill the tumor cells [42]. Our previous study showed that *Plasmodium* infection upregulates the expression levels of perforin in CD8^+^ T cells within lung cancer tissues [16]. As shown in Figure 6C,D, the PI+RT group had the highest perforin^+^CD8^+^ T/CD45^+^ cell proportion and perforin^+^CD8^+^ T cell absolute number within the tumors, among all of the groups (all *p* < 0.05). Accordingly, the PI group and RT group had a higher perforin^+^CD8^+^ T/CD45^+^ cell proportion (Figure 6C), and a higher perforin^+^ CD8^+^ T cell absolute number (Figure 6D) (PI was more significant than RT (both *p* < 0.05)), compared to the control group (all *p* < 0.05). These results suggested that PI was better than RT in promoting effector CD8^+^ T cells infiltration into tumor tissues, and that the combination of PI and RT exerted a synergistic effect on enhancing tumor-specific immune response.

Tregs [43] and MDSCs [44] are the main immune suppressor cell populations in tumors. Our results showed that the PI group had a lower Tregs/CD4^+^ T cell proportion (*p* = 0.02) and a lower MDSCs/CD45^+^ cell proportion (*p* = 0.02), while the RT group had a higher Tregs/CD4^+^ T cell proportion (*p* = 0.008) and a lower MDSCs/CD45^+^ cell proportion (*p* = 0.0003), compared to the control group (Figure 6E,F). The Tregs/CD4^+^ T cell proportion in the PI+RT group was at a level between the PI group and RT group (Figure 6E), which suggests that the down-regulation of PI on Tregs could cancel the up-regulation of RT on Tregs. The proportion of MDSCs in CD45^+^ cells in the PI+RT group was the lowest among all groups (all *p* < 0.05, Figure 6F), which suggests that the combination of PI and RT played a synergistic role in down-regulating MDSCs. MDSCs can be mainly classified into CD11b^+^Ly6C^low^Ly6G^+^ (PMN-MDSCs) and CD11b^+^Ly6C^high^Ly6G^−^ (Mo-MDSCs) subpopulations [45]. Upon further analysis of the subgroups of MDSCs, the results suggested that the combination of PI and RT down-regulated PMN-MDSCs more significantly (Appendix A).

The ratio of CD8^+^ T cells to Tregs is a marker of treatment outcome in various cancers [46,47,48,49], and the ratio of CD8^+^ T cells to MDSCs may have a similar significance. Our results indicated that the PI group had a higher ratio of perforin^+^CD8^+^ T cells to Tregs (*p* = 0.0002) and a higher ratio of perforin^+^CD8^+^ T cells to MDSCs (*p* = 0.005), and the RT group had a lower ratio of perforin^+^CD8^+^ T cells to Tregs (*p* = 0.02), but a higher ratio of perforin^+^CD8^+^ T cells to MDSCs (*p* = 0.04), compared to the control group (Figure 6G,H). The ratio of perforin^+^CD8^+^ T cells to Tregs in the PI+RT group was at a level between the PI group and RT group (both *p* < 0.01) (Figure 6G), which suggests that PI could make up for the deficiency of RT in this respect. The ratio of perforin^+^CD8^+^ T cells to MDSCs in the PI+RT group was the highest among all groups (all *p* < 0.01, Figure 6H), suggesting that the combination of PI and RT could play a synergistic role in this regard.

### 3.6. The Effect of PI and RT Combination on the Immune Profiles in the Spleens

The spleen is a major secondary lymphoid organ involved in *Plasmodium* infection [50]. We weighed the spleens of the mice and found a relationship between the spleen weights: PI > PI+RT > Con > RT (all *p* < 0.05, Appendix A), which suggests that PI increased the weight of the spleens and RT decreased the weight of the spleens.

The PI group had a higher CD45^+^ cell absolute number (*p* = 0.0003), a higher CD3^+^ T/CD45^+^ cell proportion (*p* < 0.0001), and a higher CD3^+^ T cell absolute number; and the RT group had a lower CD45^+^ cell absolute number (*p* = 0.007) and a lower CD3^+^ T cell absolute number (*p* = 0.03), but a higher CD3^+^ T/CD45^+^ cell proportion (*p* = 0.006), in comparison to the control group (Appendix A). The CD45^+^ cell absolute number (Appendix A) and CD3^+^ T cell absolute number (Appendix A) in the PI+RT group were at the levels between the PI group and RT group (all *p* < 0.05), suggesting that the defect of RT in reducing the numbers of leukocytes and T cells in the spleens could be compensated by PI. However, the CD3^+^ T/CD45^+^ cell proportion in the PI+RT group was the highest among all of the groups (all *p* < 0.05, Appendix A).

The CD8^+^ T/CD45^+^ cell proportion (all *p* < 0.05, Appendix A) and CD4^+^ T/CD45^+^ cell proportion (Appendix A) were the highest in the PI+RT group among all of the groups. The ratio of CD4^+^ T to CD8^+^ T cells in the PI+RT group was the highest among all of the groups (Appendix A). The PI group had a higher CD8^+^ T cell absolute number (*p* = 0.002) and a higher CD4^+^ T cell absolute number (*p* < 0.001), and the RT group had a lower CD8^+^ T cell absolute number (*p* = 0.04), a lower CD4^+^ T cell absolute number (*p* = 0.04), compared to the control group (Appendix A). The CD8^+^ T cell absolute number and CD4^+^ T cell absolute number in PI+RT group were both at a level between the PI and RT group (all *p* < 0.05, Appendix A).

The perforin^+^CD8^+^ T/CD45^+^ cell proportion (Appendix A) and perforin^+^CD8^+^ T cell absolute number (Appendix A) in the PI+RT group was the highest among all groups (all *p* < 0.05). The PI group had a higher perforin^+^CD8^+^ T/CD45^+^ cell proportion and a higher perforin^+^CD8^+^ T cell absolute number (both *p* < 0.0001), but the RT group had lower levels in both the proportion and absolute number (*p* = 0.05), in comparison to the control group (Appendix A). The PI+RT group had a higher perforin^+^CD8^+^ T/CD45^+^ cell proportion and a higher perforin^+^CD8^+^ T cell absolute number compared with those of the PI group or RT group (all *p* < 0.05, Appendix A). These results suggested that the combination of PI and RT significantly increased the storage of perforin^+^CD8^+^ T cells in the spleens, which is the primary anticancer force.

All of the treatment (PI, RT and PI+RT) groups had a lower Tregs/CD4^+^ T cell proportion (Appendix A) and a lower MDSCs/CD45^+^ cell proportion (Appendix A) in the spleens in comparison to the control group (all *p* < 0.05). When further analyzing the subgroups of MDSCs in the spleens, we found that all of the treatment (PI, RT and PI+RT) groups had a lower level of PMN-MDSCs (Appendix A), but not Mo-MDSCs (Appendix A), compared to the control group.

The ratio of perforin^+^CD8^+^ T cells to Tregs (Appendix A) and the ratio of perforin^+^CD8^+^ T cells to MDSCs (Appendix A) in the PI+RT group was the highest among all of the groups (all *p* < 0.05). The PI group had a higher ratio of perforin^+^CD8^+^ T cells/Tregs (*p* < 0.0001) and a higher ratio of perforin^+^CD8^+^ T cells to MDSCs (*p* < 0.0001), while the RT group had a lower ratio of perforin^+^CD8^+^ T cells to Tregs (*p* = 0.007) and a higher ratio of perforin^+^CD8^+^ T cells to MDSCs (*p* = 0.03), compared to the control group (Appendix A).

### 3.7. The Effect of PI and RT Combination on the Peripheral Blood

T cells in blood circulation are exquisitely sensitive to radiotherapy [51]. *Plasmodium* infection caused the parasitemia in the blood (Appendix A). The PI group (*p* = 0.02) and RT group (*p* = 0.009) both had a lower CD45^+^ cell absolute number than the control group (Appendix A). The PI+RT group had the lowest CD45^+^ cell absolute number among all of the groups (all *p* < 0.05, Appendix A). The PI group (*p* = 0.002) and RT group (*p* = 0.0002) both had a higher CD3^+^ T/CD45^+^ cell proportion than that of the control group (Appendix A). The PI+RT group had a higher CD3^+^ T/CD45^+^ cell proportion than that of the PI group (*p* = 0.002) and the RT group (*p* < 0.0001) (Appendix A). The RT group had the lowest CD3^+^ T cell absolute number, and the PI+RT group had a number between that of the PI group and RT group (Appendix A).

The PI+RT group had the highest CD8^+^ T/CD45^+^ cell proportion (Appendix A) and CD4^+^ T/CD45^+^ cell proportion (Appendix A) among all of the groups. The PI group and RT group both had a higher CD8^+^ T/CD45^+^ cell proportion (Appendix A) and a higher CD4^+^ T/CD45^+^ cell proportion (Appendix A) than those of the control group (all *p* < 0.05). The CD4^+^ T/CD8^+^ T cell ratio in the PI group and PI+RT group was significantly lower than that of the RT group and the control group (all *p* < 0.05, Appendix A). There was no significant difference in the CD4^+^ T/CD8^+^ T cell ratio between the RT group and the control group (Appendix A). The PI group had a higher CD8^+^ T cell absolute number (*p* = 0.04) and a lower CD4^+^ T cell absolute number (*p* = 0.04), and the RT group had a lower CD8^+^ T and CD4^+^ T cell absolute number (both *p* = 0.02) compared to the control group (Appendix A). The CD8^+^ T cell absolute number in the PI+RT group was at a level between that of the PI group and RT group (all *p* < 0.05, Appendix A). The CD4^+^ T cell absolute number in the PI+RT group was the lowest among groups (Appendix A).

The PI group had a lower Tregs/CD4^+^ T cell proportion (*p* = 0.02) and a lower MDSCs/CD45^+^ cell proportion (*p* = 0.002), and the RT group had a higher Tregs/CD4^+^ T cell proportion (*p* = 0.005) and a lower MDSCs/CD45^+^ cell proportion (*p* = 0.03), in comparison to the control group (Appendix A). The Tregs/CD4^+^ T cell proportion in the PI+RT group was at a level between that of the PI group and RT group (Appendix A), suggesting that the defect of RT in increasing this proportion was compensated by PI. The PI+RT group had the lowest MDSCs/CD45^+^ cell proportion among all of the groups (Appendix A), suggesting that the combination of both therapies played a synergistic role in this regard. In the further analysis of the subgroups of MDSCs, the results suggested that the combination of PI and RT down-regulated the PMN-MDSCs more significantly than Mo-MDSCs in the peripheral blood (Appendix A).

The PI group had a higher ratio of CD8^+^ T cells to Tregs (*p* = 0.004) and a higher ratio of CD8^+^ T cells to MDSCs (*p* = 0.005), and the RT group had a lower ratio of CD8^+^ T cells/Tregs (*p* = 0.03) and a higher ratio of CD8^+^ T cells to MDSCs (*p* = 0.007), compared to the control group (Appendix A). The ratio of CD8^+^ T cells to Tregs in the PI+RT group was at a level between the PI group and RT group (all *p* < 0.05, Appendix A), suggesting that the defect of RT in reducing this ratio was compensated by PI. The ratio of CD8^+^ T cells to MDSCs in the PI+RT group was the highest among all of the groups (all *p* < 0.05, Appendix A), suggesting that the combination of both therapies played a synergistic role in this respect.

These results suggest that the immune profiles in the spleens, peripheral blood and tumor tissues were generally similar, but there were also some differences, which might be related to the redistribution of immune cells caused by the treatments. For an antitumor immune response, the immune profiles within tumor tissues are the most important.

## 4. Discussion

For the first time, we observed a clear therapeutic effect of PI on GL261 glioma (a cold tumor), inoculated either orthotopically or subcutaneously, significantly inhibiting tumor growth and prolonging the life span of tumor-bearing mice (Appendix A). In contrast, there is no clear evidence to show that PD-1 antibody monotherapy is effective against orthotopic GL261 glioma [52]. We then combined PI with RT (the latter is known to be effective against GL261 glioma [53]), and found that this combination therapy had a significant synergistic antitumor effect, which was significantly better than each monotherapy. Importantly, the combination therapy cured GL261 glioma in approximately 70 percent of the mice (Figure 1D). The cured mice were observed for 210 days without recurrence. These cured mice were then re-inoculated with tumor cells. The results showed that the same type of tumor cells (GL261) did not develop tumors, while the different type of tumor cells (LLC) became tumors, indicating that the cured mice developed tumor-specific immune memory (Figure 1E,F). These experimental results were the same across the orthotopic model (Figure 1) and the subcutaneous model (Figure 2). We then performed a similar experiment with a model of non-small cell lung cancer (NSCLC), that is, a mouse LLC (a hot tumor) model [35]. The results were similar to those mentioned above: both PI and RT monotherapies were effective against LLC, and the combination of PI and RT also had a synergistic effect, which was significantly better than either monotherapy (Figure 3). However, the combination of both did not cure LLC in mice. We obtained lung cancer tissues for tests of cell proliferation (expression of Ki67) and cell apoptosis (ratio of cleaved caspase 3/pro-caspase 3), and the results suggested that both PI and RT could inhibit the proliferation and promote the apoptosis of tumor cells (Figure 4). The combination of both had a synergistic effect that was more significant than any single therapy (Figure 4).

Next, we analyzed the immune profiles with tumor tissues. The results showed that both PI and RT could promote the infiltration of CD3^+^ T cells into the tumor tissues of orthotopic GL261 glioma, suggesting that the cold tumor could be transformed into a hot tumor, and the combination of both had a synergistic effect, which was more significant than that of any single therapy (Figure 5). As a result, the tumor of GL261 glioma in the combination group was too small, and approximately 70% of the tumor-bearing mice (as mentioned above, regardless of orthotopic or subcutaneous model) were cured; we were unable to obtain enough samples for systemic analysis of immunophenotypes (with the exception of the simple count of CD3^+^ T cells, as mentioned above). Therefore, we could only use lung cancer specimens for a series of immune assays. The results based on lung cancer tissues showed that RT down-regulated the number of CD45^+^ leukocytes, while PI was able to compensate for this defect of RT (Appendix A). Both PI and RT significantly up-regulated the proportion of CD8^+^ T cells in CD45^+^ leukocytes, and the combination of both had a synergistic effect (Figure 6A). The combination treatment group had the highest number of CD8^+^ T cells among all of the groups (Figure 6B). For effector CD8^+^ T cells expressing perforin, both PI and RT up-regulated its proportion and number (PI was more significant than RT), and the combination of both had a synergistic effect (Figure 6C,D). In addition, we also found that PI significantly down-regulated the proportion of Tregs in CD4^+^ T cells, while RT significantly up-regulated this proportion (Figure 6E). The combination of both compensated for this defect of RT, resulting in a significantly decreased proportion in the combination therapy group, compared with the control group (Figure 6E). Both PI and RT down-regulated the proportion of MDSCs in CD45^+^ leukocytes, and the combination of both had a synergistic effect (Figure 6F). We observed that PI significantly up-regulated the ratio of perforin-expressing effector CD8^+^ T cells to Tregs, and, thus, was able to compensate for the deficiency of RT that down-regulated this ratio; the ratio in the combined treatment group was therefore significantly higher than that in the control group (Figure 6G). Both PI and RT up-regulated the ratio of perforin-expressing effector CD8^+^ T cells to MDSCs, and the combination of both was synergistic (Figure 6H). These immunophenotypic results based on tumor tissues have demonstrated that the combination of PI and RT significantly enhances the antitumor immune response in tumor-bearing mice, showing a synergistic or complementary effect. In addition to directly killing tumor cells by interrupting DNA of tumor cells, RT also enhances the efficacy of immunotherapy by inducing immunogenic death of tumor cells [54].

The trend in cancer treatment is toward combination therapy, including immunotherapy in combination with another type of immunotherapy, or combination with other anticancer therapies [55,56,57]. There is sufficient evidence to show that the combination of ICB and RT provides synergistic effects without significantly enhanced combined toxicity [58,59]. However, ICB has two disadvantages compared to PI: (1) It is ineffective against cold tumors, a condition known as primary drug resistance [60]; (2) Although it is effective against hot tumors, drug resistance may develop after a period of treatment, which is called acquired drug resistance [60]. Acquired resistance occurs because the reactivated effector T cells in tumor tissues secrete the effector molecule IFN-γ after ICB treatment. IFN-γ, on one hand, kills cancer cells; on the other hand, however, it causes residual cancer cells to express more PD-L1 [61,62,63]. At the same time, effector T cells would express more PD-1 due to the stimulation of this cytokine. In addition, IFN-γ is required for the development of immune suppressor cells, such as Tregs and MDSCs, in tumor tissues [64]. Therefore, an increase in the amount of IFN-γ in tumor tissues leads to an increase in the number of immune suppressor cells, and the levels of effector molecules, such as IL-10 and TGF-β, secreted by these cells also increase. The increase in these immunosuppressive cytokines, in turn, further up-regulates the number and function of immune suppressor cells, thus creating a vicious cycle that eventually leads to cancer cells becoming resistant to ICB [13,65,66,67,68]. However, PI is different; it down-regulates the expression level of PD-1 on effector T cells in tumor tissues (even though it up-regulates PD-1 level of T cells in peripheral blood [17]), and these T cells express and secrete perforin and granzyme B, instead of IFN-γ. Most importantly, PI down-regulates the number of Tregs, MDSCs, TAMs and CAFs, and inhibits their functions within the tumor [13,16,18]. Therefore, it can systematically relieve the tumor immunosuppressive microenvironment without producing the above vicious cycle [13,16]. Thus, there is no acquired resistance with PI treatment. In the current study, RT was observed to significantly up-regulate the proportion of Tregs in CD4^+^ T cells in tumor tissues (Figure 6E), and significantly down-regulate the number of CD45^+^ leukocytes in tumor tissues (Appendix A) and the spleen (Appendix A). ICB may not be able to compensate for these defects of RT, however, PI can produce complementary effects. In addition to facilitating the entry of effector T cells into tumor tissues, PI also promotes the entry of other immune cells, such as CD45^+^ leukocytes (observed in the current study), including NK cells and DCs (observed in our previous study [15], into tumor tissues. Therefore, PI can transform cold tumors into hot tumors, which is more effective than RT, as shown in the current study, and is thus also effective against cold tumors. Therefore, its indications should be wider than ICB. Based on these results, we proposed that PI should be more suitable than ICB as a basic treatment in combination therapy. However, as mentioned before, the specific killing ability of PI on tumor cells may be relatively weak, and RT has the advantage in this respect, so the combination of PI and RT could produce synergistic and complementary effects.

An important issue that needs to be discussed and clarified is whether *Plasmodium* infection activates or suppresses the immune system. In the field of pure malaria immunology, a series of studies have shown that *Plasmodium* infection activates the immune systems in non-tumor-bearing hosts (including animals and humans) [13,69,70]. However, there is also some research to suggest that *Plasmodium* infection suppresses the immune system, which is basically focused on the suppression of DCs or macrophages by the infection or the action of malaria pigment hemozoin. With the progression of research, it is now believed that high density parasitemia or high concentrations of hemozoin inhibits DCs/macrophages, while low level parasitemia or low concentrations of hemozoin activates DCs/macrophages [71,72,73,74]. Some studies have also shown that *Plasmodium* infection induces an increase in the number of MDSCs or Tregs in peripheral blood; therefore, it is believed that *Plasmodium* infection may suppress immune response [75,76]. However, our present study showed that *Plasmodium* infection down-regulated the proportion of MDSCs and Tregs in the peripheral blood (Appendix A) and tumor tissues (Figure 6E,F) of tumor-bearing mice, and similar findings are also found in our previous study [16]. Research also indicates that *Plasmodium* infection upregulates the expression of PD-1 and CTLA-4 on the surface of T cells in peripheral blood [77]. Nevertheless, our previous study demonstrates that both costimulatory (CD40L and GITR) and coinhibitory (PD-1, CTLA-4, TIM-3, LAG3) molecules are simultaneously expressed on CD8^+^ T cells in the peripheral blood of tumor-bearing mice infected with a *Plasmodium* parasite, without affecting the secretion of effector molecules perforin and granzyme B by these T cells [17]. These results suggest that, after the activation of the immune system by *Plasmodium* infection, the immune system itself initiates the immune balancing mechanism to prevent overactive immune response. For example, in tumor-bearing mice infected with a *Plasmodium* parasite, the PD-1 levels on CD8^+^ T cells in peripheral blood increase [17], but the levels of this molecule on CD8^+^ T cells in tumor tissues decrease significantly [16]. This suggests that a high expression of PD-1 in peripheral blood can avoid excessive systemic immune response, while a low expression of PD-1 in tumor tissues is conducive to antitumor immune response, not to mention that *Plasmodium* infection also undoes the immunosuppressive microenvironment of tumor. Therefore, based on the principle of immune equilibrium (balance) [78], the observed increase in MDSCs/Tregs and the simultaneous expression of coinhibitory molecules on T cells in peripheral blood cannot be attributed to the suppression of the immune system by the *Plasmodium* infection itself; instead, they can serve as the markers of immune activation. Interestingly, our recent study indicates that subsequent *Plasmodium* infection induces a high proportion of CD4^+^CD28^high^CD95^high^ central memory T cells and a strong SIV (simian immunodeficiency virus)-specific T cell response drives the hosts to maintain the diversity of SIV-specific T cell receptor repertoire, generating new SIV-specific T cell clones to track the antigenic variations of SIV, and thus extending the life span of rhesus monkeys infected with SIV [79]. This suggests that *Plasmodium* infection enhances immune response to different pathogen and drives T cells to track its variations, and thus may also drive T cells in tumor-bearing hosts to trace the antigenic variations of tumor cells. In brief, our series of studies show no conflicting results that either high or low densities of parasitemia *Plasmodium* infection activates the immune system in tumor-bearing mice [15,16,17,18] [and unpublished data]. Nevertheless, based on the activation of the immune system, at least by the low grade of *Plasmodium* infection mentioned above, our clinical trial protocol of *Plasmodium* immunotherapy for advanced solid tumors requires the use of antimalarial drug (artesunate) to control parasitemia at a reasonably low level (below 0.1% or 0.05%), which has been shown to activate the immune systems of cancer patients [13]. In theory, the immune system of the tumor-bearing host has already been suppressed by cancer cells, and *Plasmodium* parasite, as a foreign pathogen, has strong danger signals, including a large amount of pathogen-associated molecular patterns (PAMPs) [69] and more than 5000 heterologous proteins [80]; therefore, the overall effect of *Plasmodium* infection on the immune system should be wake-up and activation rather than inhibition.

Furthermore, such a low-level parasitemia (but not malaria) has been shown to be safe for patients with advanced cancer in our clinical trials. For example, we have treated and observed more than 140 patients thus far, none of whom have died from the parasite infection, and none of whom have developed serious life-threatening complications (unpublished data). Perhaps one would speculate that because artemisinin-type drugs (including artesunate) can activate antitumor immune responses and have antitumor effects [81], if efficacy is observed in the clinical trials, it may be due to the effect of artesunate rather than the effect of *Plasmodium* immunotherapy. However, we only use 6–12 mg artesunate each time, and each patient requires this only one to four times, thus the total dose over the course is 6–48 mg, while the dose used intravenously for malaria treatment is 60 mg/day for 5–7 days, and the total dose over the course is 300–420 mg (if administered orally, the total dose is 600 mg) [82]. Importantly, in clinical trials of artemisinin-type drugs for cancer treatment, the general dosage is 100–200 mg/day for 14–28 days, and the total dose range is 1400–5600 mg [83]. Therefore, the amount of artesunate we used to control the parasite density in *Plasmodium* immunotherapy is insignificant in eliciting an antitumor immune response. Incidentally, in the treatment of more than 140 patients, we have not observed artesunate resistance of the parasites in any of the treated patients (unpublished data). In the event that the parasite in a patient becomes resistant, we are also absolutely certain that we can completely kill the parasite at the end of the treatment, because the strain of *Plasmodium vivax* we used are sensitive to all existing antimalarial drugs, including chloroquine, quinine, and atovaquone, etc. [13]. More importantly, under the premise of ensuring clinical safety, as mentioned above, we did observe significant therapeutic effects in some patients with advanced cancer treated with *Plasmodium* immunotherapy. One patient with advanced lung adenocarcinoma has been observed for more than 5 years after treatment and is currently in tumor-free survival and has returned completely to normal life. The patient had no indication for surgery before *Plasmodium* immunotherapy and was resistant to multiple targeted drugs. After a course of *Plasmodium* immunotherapy, the enlarged lymph nodes in the neck and supraclavicular disappeared. CT examination revealed that the primary lung lesion had morphologic changes, from “crab shape” to “plaque shape”. After minimally invasive surgery, the primary lesion was removed, and macroscopic observation revealed that the tumor had been encapsulated. Pathological examination revealed a large number of immune cell infiltration in the tumor tissue, including a large number of CD3 positive T cells. This was in sharp contrast to the pathological sections (almost no immune cells) from another patient with advanced lung adenocarcinoma that had not been treated with *Plasmodium* immunotherapy (unpublished data). These results are consistent with those observed in the mouse lung cancer models.

There are some shortcomings in this study: The mouse tumor cell GL261 we used does not completely represent human glioma or glioblastoma, because GL261 still has certain immunogenicity and is not a completely “cold” tumor [84]. In addition, we are unable to obtain enough tumor tissues of GL261 glioma of the PI+RT group for systematic immunophenotypic analysis, and data mainly based on lung cancer (hot tumor) tissues may not reflect the real situation of glioma (cold tumor).

## 5. Conclusions

For the first time, we report the preclinical research results of PI combined with RT in the treatment of mouse glioma (GL261) and non-small cell lung cancer (LLC). These results show that the combination of PI and RT produces significant synergistic antitumor effects and, in particular, can cure approximately 70% of glioma, without a significantly enhanced effect of joint toxicity. The preliminary immunological mechanisms of these synergistic and complementary effects are described, and deeper immunological and molecular mechanisms need to be further explored. The clinical safety and public health security of PI alone have been preliminarily described in our recent publication [13]. Meanwhile, based on our current studies, we suggest conducting clinical trials of PI combined with RT in the treatment of solid tumors, especially glioblastoma or glioma.

## Figures and Tables

**Figure 1 cells-11-03600-f001:**
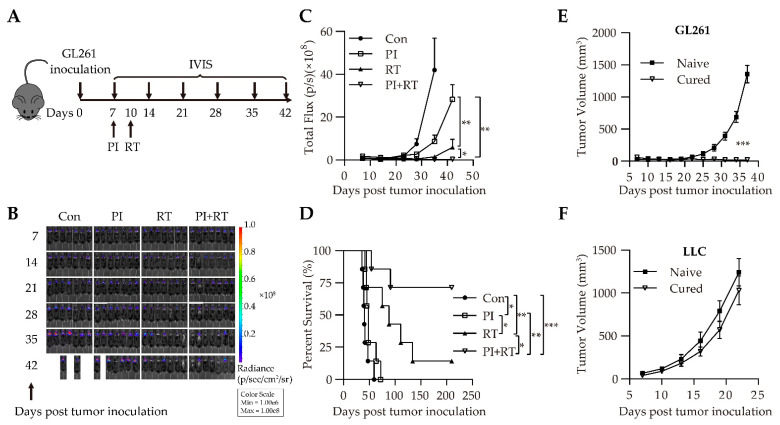
Therapeutic efficacy of PI in combination with RT in orthotopic GL261-Luc glioma model. (**A**) Schematic representation of combination therapy in orthotopic intracerebral GL261-Luc tumor-bearing mice. (**B**) Living imaging for all groups. (**C**) Tumor growth curves based on the data on days 7, 14, 21, 28, 35 and 42 after inoculation of the GL261-Luc cells (*n* = 7 per group). Photon-density heat maps were shown for all groups at the same bioluminescence signals. The statistical differences at the experiment endpoint between groups were analyzed with an unpaired two-tailed Student’s t-test. (**D**) The Kaplan-Meier survival curves of mice (*n* = 7) were shown and analyzed by a log-rank test. “Cured” mice (*n* = 6) were rechallenged with 2 × 10^6^ syngeneic GL261-Luc cells (**E**) in the right flank and non-syngeneic 5 × 10^5^ LLC cells (**F**) in the left flank. Naïve mice (*n* = 6) received the same rechallenges as control. The data showed the mean ± SEM. Statistical differences were indicated by the *p* values, *, *p* ≤ 0.05; **, *p* ≤ 0.01; ***, *p* ≤ 0.001.

**Figure 2 cells-11-03600-f002:**
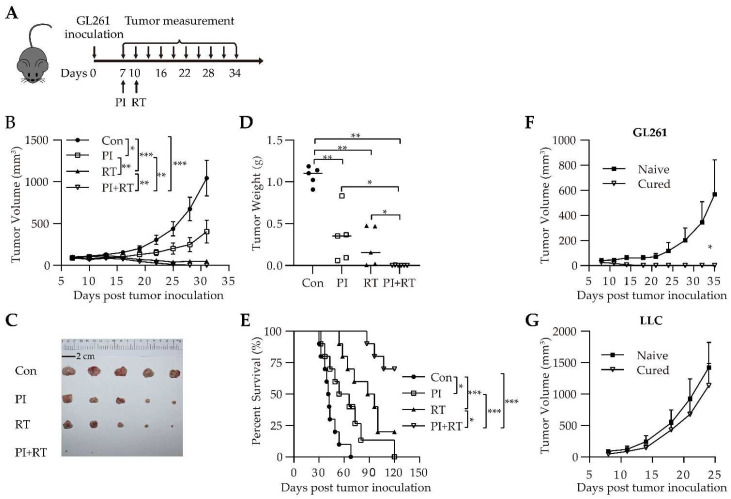
Therapeutic efficacy of PI in combination with RT in subcutaneous GL261 tumor model. (**A**) Schematic representation of combination therapy in subcutaneous GL261 tumor-bearing mice. (**B**) Tumor growth curves (*n* = 10 per group). At the endpoint, the statistical differences between groups were analyzed with an unpaired two-tailed Student’s t-test. (**C**) Tumor sizes and (**D**) weight of the s.c. tumor mass on day 29 after tumor cell inoculation (*n* = 5). The statistical differences between groups were analyzed with an unpaired two-tailed Student’s t-test. (**E**) The Kaplan-Meier survival curves of mice (*n* = 10). Survival curves were analyzed by a log-rank test. Cured mice were rechallenged with 2 × 10^6^ syngeneic GL261 cells (**F**) in the right flank and non-syngeneic 5 × 10^5^ LLC cells (**G**) in the left flank (*n* = 6) after 120 days post inoculation and compared with 6 naïve mice. The data showed the mean ± SEM. Statistical differences were indicated by the *p* values, *, *p* ≤ 0.05; **, *p* ≤ 0.01; ***, *p* ≤ 0.001.

**Figure 3 cells-11-03600-f003:**
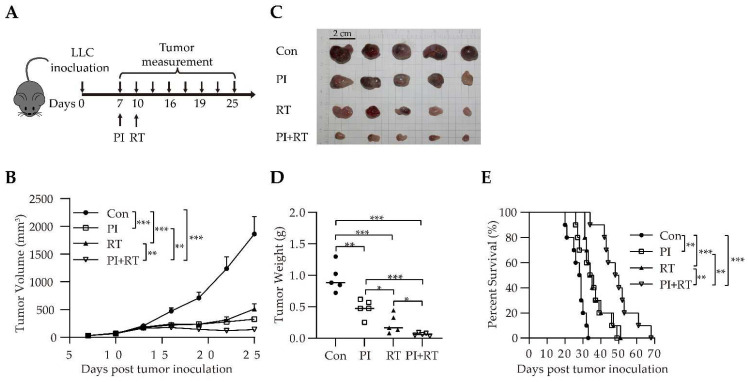
Therapeutic efficacy of PI in combination with RT in subcutaneous LLC lung cancer model. (**A**) Schematic representation of combination therapy in subcutaneous LLC tumor mice. (**B**) Tumor growth curves (*n* = 10 per group). The statistical differences between groups were analyzed with an unpaired two-tailed Student’s *t*-test. (**C**) Comparisons by visual observation of tumor sizes between groups. (**D**) Weight of the s.c. tumor mass on day 19 after tumor cell implantation (*n* = 5). The statistical differences between groups were analyzed with an unpaired two-tailed Student’s *t*-test. (**E**) The Kaplan-Meier survival curves of mice (*n* = 10). Survival curves were analyzed by a log-rank test. The data showed the mean ± SEM. Statistical differences were indicated by the *p* values, *, *p* ≤ 0.05; **, *p* ≤ 0.01; ***, *p* ≤ 0.001.

**Figure 4 cells-11-03600-f004:**
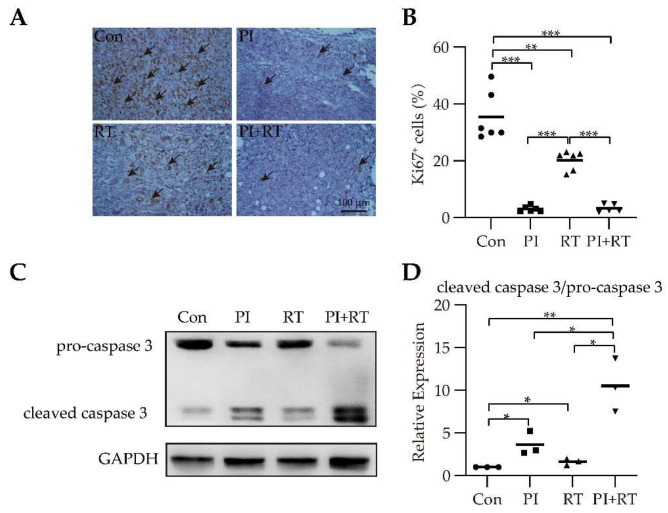
The effect of PI in combination with RT on cell proliferation and apoptosis in subcutaneous LLC lung cancer model. (**A**) Immunohistochemical staining: Arrows showed Ki67-expressing cells. (**B**) Assay quantification for Ki67 at 400× magnification (*n* = 6). (**C**) Western blotting assay results of pro-caspase 3 and cleaved caspase 3. (**D**) Assay quantification for caspase 3: Ratio of cleaved caspase 3/pro-caspase 3 (*n* = 3). GAPDH was used as a loading control for western blotting. The statistical differences between groups were analyzed with an unpaired two-tailed Student’s *t*-test. The data showed the mean ± SEM. Statistical differences were indicated by the *p* values, *, *p* ≤ 0.05; **, *p* ≤ 0.01; ***, *p* ≤ 0.001.

**Figure 5 cells-11-03600-f005:**
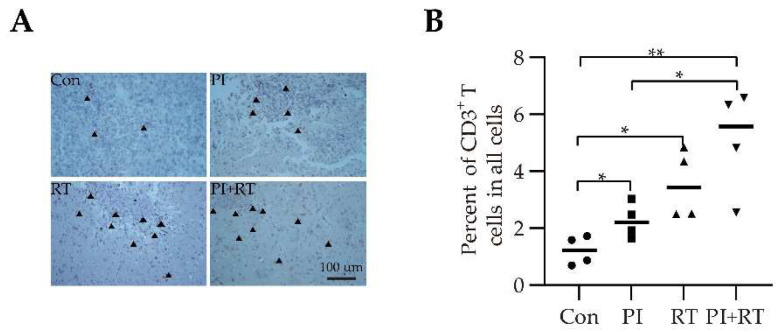
The effect of PI in combination with RT on the infiltration of CD3^+^ T cells in tumor tissues in orthotopic GL261-Luc glioma-bearing mice. (**A**) Immunohistochemical staining and quantification of CD3 at 400× magnification. Black arrows showed CD3-expressing T cells. (**B**) The percentage of CD3^+^ T cells among all calculated cells in tumor tissues (*n* = 4). The statistical differences between groups were analyzed with an unpaired two-tailed Student’s *t*-test. The data showed the mean ± SEM. Statistical differences were indicated by the *p* values, *, *p* ≤ 0.05; **, *p* ≤ 0.01.

**Figure 6 cells-11-03600-f006:**
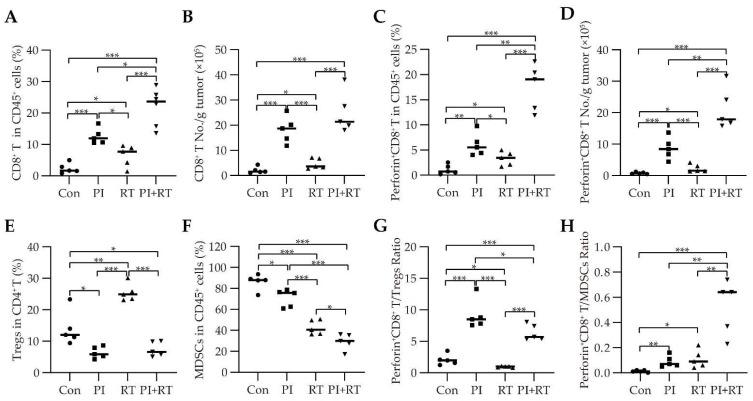
The effect of PI in combination with RT on the immune profiles in tumor tissues in subcutaneous lung cancer-bearing mice. Lymphocytes were isolated from tumors on day 19 post inoculation (*n* = 5). (**A**) Proportion of CD8^+^ T cells in CD45^+^ cells. (**B**) Absolute number of CD8^+^ T cells. (**C**) Proportion of perforin^+^CD8^+^ T cells in CD45^+^ cells. (**D**) Absolute number of perforin^+^CD8^+^ T cells. (**E**) Proportion of Tregs in CD4^+^ T cells. (**F**) Proportion of MDSCs in CD45^+^ cells. (**G**) Ratio of perforin^+^CD8^+^ T cells to Tregs. (**H**) Ratio of perforin^+^CD8^+^ T cells to MDSCs. Absolute number was presented as the number of cells per gram of tumor. The statistical differences between groups were analyzed with an unpaired two-tailed Student’s t-test. The data showed the mean ± SEM. Statistical differences were indicated by the *p* values, *, *p* ≤ 0.05; **, *p* ≤ 0.01; ***, *p* ≤ 0.001.

## Data Availability

All data presented to support the finding in the study are available in the article and Appendix A. Raw data or further inquiries can be directed to the corresponding authors, upon reasonable request.

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
