# Peer review of "Preclinical Study of Plasmodium Immunotherapy Combined with Radiotherapy for Solid Tumors"

_cells, 2022, doi:10.3390/cells11223600_

Round 1

Reviewer 1 Report

The manuscript called: “Preclinical Study of Plasmodium Immunotherapy Combined with Radiotherapy for Solid Tumors“ is highly interesting and detailed. The study shows a significant synergistic anti-tumor effect of radiotherapy and infection with Plasmodium yoelii in orthotopic and subcutaneous models of mouse glioma and subcutaneous model of mouse non-small cell lung cancer. The study is complex, describes changes in the density of tumor-infiltrating immune cells, as well as in the proportions of immune cells in the peripheral blood and spleen of treated mice. The major issue of the study is the question of safety of controlled Plasmodium infection in human patients.

Although there is a reference to a paper describing the conditions of Plasmodium immunotherapy in clinical trials, a brief summary should be also included in the discussion. Many patients in later stages of cancer do not tolerate even the ICI and Plasmodium infection with subsequent malaria treatment could be unbearable. Importantly, development of resistance to antimalaric drugs, such as artemisinins, is also possible.

An effect of artemisinin on the immune response was not described in this study. However, in clinical trials, artemisinin is used to control the parasitemia. Are there any known effects of artemisinin on the anti-tumor immune response?   

The quality of figures is unfortunately very low, but probably it is due to PDF creation. In Fig. 1B, the mice are barely visible, and it was not possible to see the effect of treatment. Similarly, the IHC pictures in Fig. 4 and 5 are of a very low quality and therefore have low validity.

Reviewer 2 Report

The study by Tao et al "Preclinical Study of Plasmodium Immunotherapy Combined with Radiotherapy for Solid Tumors" is a fascinating study. The author represented the study very extensively. But some improvements are needed in order to publish the manuscript.

1. Section 2.1: Why only female mice were selected for the study?

2. How was the terminal point for the survival analysis "cure" group determined? Did the mice didn't survive after that point or were they sacrificed?

3. Section 2.6: what was the purpose of doing xenograft after the transplantation in the striatum with glioblastoma cells? If the author wanted to find out the imprint of memory cells, how did they measure it? A detailed flow cytometric analysis for the activation of immune cells is helpful to justify.

4. Section 3.1: Cured mice didn't produce large tumors ---did the author treat the cured mice similarly as before after doing the xenograft?

5. Fig 1B is hard to visualize. Please upload a higher-resolution image

6. Author mentioned immunosuppressive cells. Did they try to target MDSCs or Tregs?

7. Did the author try to check the PD-l1 status?

8. Why glioblastoma tissue was not analyzed by the flow?

9. Lastly it would be good if the author can show some comparison with human samples. At least a paragraph in the discussion.

Reviewer 3 Report

The Authors report the preclinical research results of Plasmodium immunotherapy (PI) combined with Radiotherapy (RT) in the treatment of mouse glioma (GL261). These results show that the combination of PI and RT produces significant synergistic effects, and in particular, the combination of both therapies can cure about 70% of glioma, without significantly enhanced effect of toxicity. The preliminary immunological mechanisms of these synergistic and complementary effects are described. The clinical and public health safeties of PI have been preliminarily described in their previous publication. The manuscript has the limitations that the authors reported. The study seems to me well conducted in the complexity process to arrive at defining the conclusions. The manuscript can be accepted in the form in which it was presented.
